# Interventions to improve the mental health of women experiencing homelessness: A systematic review of the literature

**Joanna Anderson[1], Charlotte Trevella[2], Anne-Marie Burn**📙[1]*

**1** Department of Psychiatry, University of Cambridge, Cambridge, United Kingdom, **2** Department of Public Health and Primary Care, University of Cambridge, Cambridge, United Kingdom

* amb278@cam.ac.uk

## Abstract

### Background

Homelessness is a growing public health challenge in the United Kingdom and internationally, with major consequences for physical and mental health. Women represent a particularly vulnerable subgroup of the homeless population, with some evidence suggesting that they suffer worse mental health outcomes than their male counterparts. Interventions aimed at improving the lives of homeless women have the potential to enhance mental health and reduce the burden of mental illness in this population. This review synthesised the evidence on the effectiveness and acceptability of interventions which aim to improve mental health outcomes in homeless women.

### Methods

Five electronic bibliographic databases: MEDLINE, PsycInfo, CINAHL, ASSIA and EMBASE, were searched. Studies were included if they measured the effectiveness or acceptability of any intervention in improving mental health outcomes in homeless women. Study quality was assessed using the Effective Public Health Practice Project (EPHPP) Quality Assessment Tool. A narrative summary of the study findings in relation to the research questions was produced.

### Results

Thirty-nine studies met inclusion criteria. Overall, there was moderate evidence of the effectiveness of interventions in improving mental health outcomes in homeless women, both immediately post-intervention and at later follow-up. The strongest evidence was for the effectiveness of psychotherapy interventions. There was also evidence that homeless women find interventions aimed at improving mental health outcomes acceptable and helpful.

### Conclusions

Heterogeneity in intervention and study methodology limits the ability to draw definitive conclusions about the extent to which different categories of intervention improve mental health

**Data Availability Statement:** All relevant data are within the paper and its Supporting information files.

**Funding:** The author(s) received no specific funding for this work.

outcomes in homeless women. Future research should focus on lesser-studied intervention categories, subgroups of homeless women and mental health outcomes. More in-depth qualitative research of factors that enhance or diminish the acceptability of mental health interventions to homeless women is also required.

**Competing interests:** The authors have declared that no competing interests exist.

## Introduction

Homelessness is a growing public health problem in the United Kingdom and internationally [1–3]. In England, 76,860 households were assessed as homeless, or threatened with homelessness, in the third quarter of 2022, an increase of 4.4% from Quarter Three of the previous year [1]. Defining homelessness can be challenging, and there is no international consensus on a definition [3]. However, increasingly, organisations are adopting broader conceptions of homelessness, which include insecure and inadequate housing [4–6]. The United Nations Economic and Social Council defines homelessness as 'a condition where a person or household lacks habitable space with security of tenure, rights and ability to enjoy social relations, including safety' [4].

Women represent a vulnerable subgroup of the homeless population with distinct health and social needs [7–9]. Research into this group has been neglected historically, with most early studies of homelessness focused on men sleeping rough [10]. This is partly because women frequently experience less visible forms of homelessness; for example, living in precarious arrangements with family or friends, or occupying temporary accommodation, domestic violence shelters or temporary housing [11]. These informal arrangements may be unsafe and exploitative, such as the exchange of sex for accommodation, or residing with abusive partners [7,12]. Women who do choose to sleep on the streets often hide to protect themselves from violence and are therefore less likely to be included in counts of rough sleepers (i.e. people who are homeless and sleep outside or in places not suitable to live in, such as abandoned buildings or under bridges) [13]. Moreover, women-centred homeless services are absent or inadequate in most developed countries [9,14]. As a result, female homelessness is often concealed, however, it is not uncommon. In Quarter 3 of 2022, 88% of statutorily homeless single parent households with dependent children were headed by a woman, as well as 33% of single person households [1].

There is extensive evidence that homeless populations suffer poor mental health outcomes [15]. Mental illness can be both a cause and a consequence of homelessness [16]. The same risk factors which contribute to homelessness, such as childhood trauma, exposure to violence, and poverty, also predispose an individual to mental illness [17,18]. Homeless people are often separated from their usual support networks and exposed to stigma and marginalisation, exacerbating the risk further [19].

Rates of depression, post-traumatic stress disorder (PTSD), and substance misuse are particularly high among homeless women [20–22]. Estimates of the prevalence of mental illness in homeless women are variable, ranging from 48% to 85% for all mental disorders [23,24], and 16% to 82% for alcohol or drug misuse [23,24]. Some studies indicate that homeless women are also more likely to suffer from mental illness, particularly depression and PTSD, than their male counterparts [24–26]. This may be partly explained by the high proportion of homeless women who report histories of domestic violence and sexual abuse [26–29]. Such distressing life events may predispose to mental health problems, especially under the challenging conditions of homelessness.

As well as causing individual distress, mental illness and substance misuse compromise the ability of homeless women to apply for housing and financial assistance, obtain employment and seek social support, thus prolonging the period of homelessness [21]. Mental illness may also compromise the ability of homeless women to foster supportive and loving relationships with their children, which can result in attachment disorders [16,23] and generational transmission of trauma [16,30].

Homeless women have different characteristics and vulnerabilities to homeless men and may experience worse mental health outcomes than their male counterparts [9,31–34]. However, there are relatively few studies of interventions specifically targeted at improving mental health outcomes in homeless women [1]. This paper is the first known systematic review to synthesise the evidence on effectiveness and acceptability of interventions aimed at improving mental health outcomes in homeless women.

### Aims and research questions

This systematic review of the literature aims to appraise and synthesise evidence, and answer the following research questions:

1. *Which interventions for homeless women are most effective in improving mental health outcomes in this population*?

2. *What is the acceptability to study participants of interventions to improve mental health outcomes in homeless women*?

## Methods

The review was prospectively registered with PROSPERO (registration number: CRD42022307588) on 2nd February 2022 and is reported according to the Preferred Reporting Items for Systematic Reviews and Meta-Analyses (PRISMA) Guidelines [35].

### Search strategy and selection criteria

Relevant studies were identified by systematic keyword searching of the following electronic databases: MEDLINE, EMBASE, PsycINFO, ASSIA and CINAHL. Supplementary searches were conducted by forward and backward citation searching of included studies and related systematic reviews. A combination of terms for homelessness, women, interventions, and mental health were used in the search strategy. For the full search strategy, see S1 File. Initial searches were conducted on 21st February 2022 and updated searches were completed on 14th May 2023. To ensure systematic and consistent process of study selection inclusion criteria shown in Table 1 were developed using the Population, Intervention, Comparison, Outcome (PICO) Framework [36].

### Study selection

Search results were managed using the bibliographic software CADIMA (https://www.cadima.info/index.php). Results of searches from different databases were merged and duplicated studies removed. Titles and abstracts for all studies generated by the search strategy were reviewed and obviously irrelevant studies rejected. The full texts of remaining studies were obtained, then full texts were read and reviewed against inclusion and exclusion criteria. At both screening stages, one author (CT) screened all records, and two other authors (AMB, JA) independently reviewed 10% of articles each. The agreement between raters in title/abstract

**Table 1. Inclusion criteria.**

| Criteria | Inclusion criteria |
|---|---|
| Population | Populations in which >90% identify as girls or women and are homeless or experiencing housing exclusion according to the UN Economic and Social Council definition of homelessness [4] at the time of recruitment into the study. No age restrictions are applied. |
| Intervention | Any intervention that is targeted at this population, and has been developed, modified or introduced with the intention of improving mental health outcomes (+/- other outcomes) |
| Comparator | • Study participants prior to their exposure to the intervention<br>• Study participants assigned to a different intervention<br>• Study participants assigned to no intervention / care as usual |
| Outcomes | Question One:<br>• Studies which include validated measures of symptoms or severity of specific mental disorders named in the DSM-V as primary or secondary outcomes<br>• Studies which include validated measures of psychological distress, overall psychiatric symptom severity, or overall mental health status e.g. the mental health subscale of SF-36, SCL-90-R, the psychological domain of WHO-QOL scale<br>Question Two:<br>Acceptability of the intervention measured by qualitative or quantitative data collection. Acceptability includes measures of:<br>• Client satisfaction<br>• Perceived usefulness<br>• Attitudes to treatment<br>• Perceived barriers to the intervention working |
| Setting | Any setting in which an intervention can be delivered to homeless women. This could include hospitals or healthcare centres, community centres, public spaces, shelters, hostels, or temporary accommodation. |
| Study design | • Randomised controlled trials (RCTs) including wait-list RCTs, matched-pair RCTs, cluster RCTs<br>• Non-randomised trials<br>• Before-and-after studies<br>• Prospective cohort studies<br>• Qualitative studies (for Question Two) |
| Country | All high- and middle-income countries |
| Date | From the earliest available |
| Language | English |

screening stage was $\hat{k} = 0.68$ and in full text screening $\hat{k} = 0.84$. Any differences resolved with discussion.

## Data extraction

Data from selected studies was extracted and presented in a piloted spreadsheet. Intervention data was extracted using The Template for Intervention Description and Replication (TIDieR) Reporting Checklist [37] which included details on the rationale, materials, procedures, providers, mode of delivery, location, frequency and quantity of each intervention. Detailed extraction tables are included in S2 File.

## Critical appraisal

The quality of quantitative studies was appraised using the Effective Public Health Practice Project (EPHPP) Quality Assessment Tool for Quantitative Studies [38]. For qualitative studies, study quality was appraised using the Critical Appraisal Skills Programme (CASP) Qualitative Research Appraisal Checklist [39]. CASP tool comprises of three sections apprising

validity of results (including clarity of research aims, appropriateness of methodology, recruitment and study conduct); quality of a study (including ethical considerations, data analysis method, statement of findings; and applicability of results (including contribution to existing knowledge, identifying new areas of inquiry and transferability of findings). One author (CT) rated every study, and two authors (AMB, JA) independently checked quality appraisals for 10% of included papers. Any disagreements were resolved by discussion.

### Data synthesis

Due to high levels of heterogeneity in terms of study designs, nature of interventions and outcome measures, it was inappropriate to conduct a meta-analysis as part of this systematic review. Instead, narrative synthesis was conducted to summarise evidence, form conclusions, and develop recommendations regarding policy and practice, using the framework developed by Popay et al. [40]. This framework was applied separately to each research question.

## Results

### Study selection

Thirty-nine studies met the inclusion criteria. A PRISMA flowchart detailing the identification of the studies is given in Fig 1.

### Study characteristics

Of the included papers, 30 related to the first review question, three related to the secondary review question, and seven related to both questions. For three unique interventions, earlier studies served as pilots for later studies of the same intervention, in some cases with minor modifications to the intervention based on results of the pilot studies [41–49]. General characteristics of included studies are reported in Table 2.

The 39 studies were published between 1998 and 2023, with the majority of studies published between 2013 and 2023. All studies were peer-reviewed journal articles. The majority of the studies were conducted in the USA (n = 27). The other countries represented were South Africa, Canada, Mexico, South Korea, The Netherlands and Spain. All studies were conducted in urban settings. Twenty studies were randomised controlled trials (RCTs), and the remaining studies were a mixture of controlled (n = 12) and uncontrolled (n = 5) before-and-after studies, and qualitative studies (n = 2). Of the 37 quantitative studies, 32 used a control group as the comparator, and five used the pre-intervention period as the comparator. The median number of time points was three (range = 10), and median duration of follow-up was 26 weeks (range = 79 weeks). Therefore, there was significant variation across the studies in number of time points and duration of follow-up. Detailed characteristics of individual studies are reported S3 File.

### Participant characteristics

Studies included in this review focused on different subgroups of homeless women; there were no studies that included both male and female participants. Some studies' inclusion criteria restricted participation to homeless women who were also mothers (n = 12), domestic violence victims (n = 13), young women (n = 7), veterans (n = 2), elderly women (n = 1), or parolees or probationers (n = 1). Some studies included only homeless women who met diagnostic criteria for a psychiatric disorder (n = 2) or a substance use disorder (n = 6). The median number of participants in included studies was 68.5, with a range of 14 to 653 participants. The mean age of participants was 34.5 with a range of 16 to 72. Ethnicity was recorded in 28 studies; off

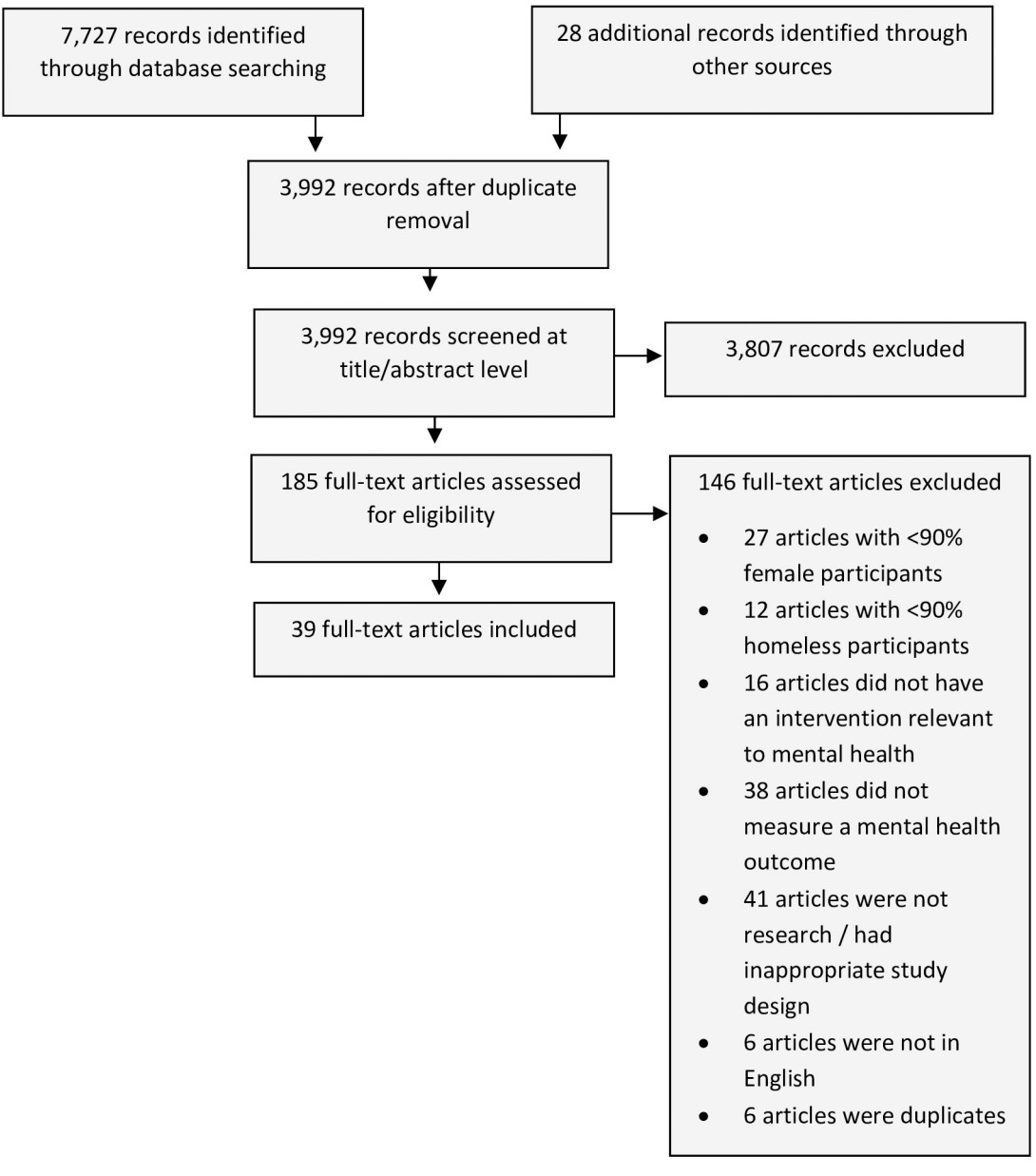

**Fig 1. PRISMA chart.**

these, 20 studies reported that the majority of participants were Black, five studies reported that the majority of participants were White, the remaining two studies reported that the majority of participants were Hispanic, or minority not otherwise specified.

## Intervention characteristics

The duration of interventions ranged from five days to one year. Interventions were delivered by psychotherapists; peer mentors; nurses; case managers; shelter staff; community health workers; residential facility managers; primary care providers and members of research teams. Delivery settings included shelters; residential treatment centres; primary

**Table 2. General characteristics of included studies.**

| Criteria | Characteristics | Number of studies |
|---|---|---|
| Year | 1991–2000 | 1 |
| | 2001–2010 | 9 |
| | 2011 onwards | 29 |
| Country | USA | 27 |
| | South Africa | 1 |
| | Canada | 3 |
| | Mexico | 1 |
| | South Korea | 2 |
| | The Netherlands | 1 |
| | Spain | 4 |
| Study design | RCT | 20 |
| | Controlled before-and-after study | 12 |
| | Uncontrolled before-and-after study | 5 |
| | Qualitative study | 2 |
| Comparator | Control group | 32 |
| | Pre-intervention period | 5 |
| | No comparator (qualitative study) | 2 |
| Outcomes [a] | Depression | 19 |
| | Anxiety | 9 |
| | PTSD | 8 |
| | Substance use | 15 |
| | General psychological health status | 17 |
| | Acceptability | 11 |
| Outcome measurement time points | One (qualitative study) | 2 |
| | Two (baseline and post-intervention) | 10 |
| | Three | 11 |
| | Four | 10 |
| | Five | 3 |
| | Six or more | 2 |
| Duration of follow-up (post-intake) | Qualitative with cross-sectional data collection | 2 |
| | 1 week | 2 |
| | 2 weeks | 1 |
| | 2 months | 4 |
| | 3 months | 3 |
| | 5 months | 1 |
| | 6 months | 11 |
| | 9 months | 3 |
| | 12 months | 7 |
| | 15 months | 1 |
| | 18 months | 2 |
| | 24 months | 1 |

[a]Some studies measured more than one outcome, so total does not add to 39.

care practices; community organisations; and the houses of formerly homeless women who had secured housing. There was a high degree of diversity in the interventions implemented across studies. The interventions in the included studies can be divided into the following categories:

**Psychotherapy interventions.**    Eleven studies of seven unique interventions examined the effectiveness of psychotherapy interventions in improving mental health outcomes in homeless women. In all studies psychotherapy was delivered in person, whether in a group context (n = 6), as individual sessions (n = 4) [43–46] or a mixture of group and individual sessions (n = 1) [50]. The most commonly used therapeutic model was cognitive behavioural therapy (CBT) [43–46,50–52]; in addition dialectical behavioural therapy (DBT) [53], resilience enhancement therapy [54], and crisis intervention [55] were used.

**Multifactorial interventions.**    Twelve studies of 10 unique interventions measured the outcomes of multifactorial interventions. In the studies in this review, multifactorial interventions combined two or more of the following components: housing, psychotherapy, case management, parenting skills, relaxation and recreation, health education and social support [41,42,47–49,56–62].

**Social support interventions.**    There were three studies of two unique social support interventions. One of these interventions was based on a group social support model [63], the other intervention had peer mentor and group elements to its delivery [48,49].

**Recreation and relaxation interventions.**    Three studies were of recreation and relaxation interventions [60,64,65]. One of these studies used a combination of exercise and meditation [60], another involved progressive muscle relaxation to music [56], while a third was a gardening intervention [64].

**Case management interventions.**    The two studies in the case management category were based on adaptions of the same case management protocol: Critical Time Intervention (CTI) [66,67], a nine-month intervention designed to enable case managers to facilitate periods of transition from shelters to community living [66].

**Collaborative care model interventions.**    Two studies investigated the effect of collaborative care model interventions upon mental health outcomes [68,69]. The collaborative care model constitutes team-based care, with an emphasis on care coordination and management, and regular and proactive monitoring of selected health outcomes [70]. In these two studies the model was adapted to support the management of substance misuse and depression, respectively.

**Parenting skills interventions.**    Three studies examined the effect of parenting skills interventions on the mental health of homeless mothers [71,72].

A detailed summary of the characteristics of all interventions is provided in Table 3.

## Study quality

Of the 37 quantitative studies included, according to the EPHPP Quality Assessment Tool, 18 studies were rated strong, 14 studies were rated moderate, and five studies were rated weak. The relatively high proportion of high-quality studies is due to the large number of RCTs included. Common sources of bias identified across studies were failure to adequately control for confounding variables and bias due to loss to follow-up. There were a large number of smaller studies, 12 studies had fewer than 50 participants and six studies had fewer than 20 participants. Of the two qualitative studies included, one scored 6/10 and a second scored 8/10 on the CASP checklist, indicating moderate to high quality. The quality of included quantitative studies is summarised in Table 4.

## Effectiveness of interventions for homeless women in reducing depression

**Psychotherapy interventions.**    Nine studies measured the effect of psychotherapy interventions on depression severity. Results generally indicated that psychotherapy is effective at

**Table 3. Characteristics of interventions.**

| Study | Name of intervention | Goal of intervention | Materials and procedures | Provider | Mode of delivery | Location delivered | Frequency and duration | Fidelity |
|---|---|---|---|---|---|---|---|---|
| **Bain, 2014** | New Beginnings | Improve homeless mothers' mental health, sensitivity to their infants' needs, and reflective function, and enhance infants' development | Group psychotherapy sessions to explore experiences of motherhood, improve emotional regulation and build empathy between mother and infant | Four volunteer psychologists with training in the programme | Face-to-face in groups with 6–9 mother-infant dyads and one psychologist | Homeless shelters | 12 sessions of 1.5 hour duration | Weekly supervision meetings with psychologists to check programme adherence |
| **Bani-Fatemi et al., 2020 Kahan et al., 2020** | Peer Education and Connection through Empowerment (PEACE) | Empower and support homeless female survivors of domestic violence | Trauma-informed group psychoeducation sessions to discuss topics such as identity formation, women's health, relationships, coping mechanisms. Group social and skill-based activities e.g. yoga, crafts, cooking | Paid peer mentors who received 12 hours of training | Face-to-face groups with eight participants and two peer mentors | Community resource centre and shelter for homeless youth | 16 once-weekly sessions | Not monitored |
| **Castaños-Cervantes, 2019** | Brief group Cognitive Behavioural Therapy (CBT) | Improve subjective wellbeing, symptoms of anxiety and depression, assertive behaviours and emotional regulation in homeless girls | Group CBT sessions to learn emotional regulation and behavioural management. CBT adapted to be age- and culturally-appropriate e.g. with use of audio-visual materials and craft activities | Psychologists | Face-to-face groups with six participants per group | Classrooms at collaborating institutions | Eight once-weekly three-hour sessions | Not monitored |
| **Constantino et al., 2005** | Social Support Intervention | Improve social support and overall wellbeing and reduce psychological distress among homeless women in domestic violence shelters | Group sessions to develop positive relationships between women at the shelter, and to foster a sense of belonging, self-awareness, self-esteem and knowledge of available supports in the community | Trained nurses | Face-to-face groups | Domestic violence shelter | Eight once-weekly 1.5-hour sessions | Not monitored |
| **Desai et al., 2008** | Seeking Safety | Improve mental health and social functioning of homeless female veterans with a mental health or substance misuse problem | Individual or group CBT sessions addressing issues such as safe behaviours, healthy relationships, life skills and relapse prevention | Case managers who had received training and assessment in programme delivery | Face-to-face group and individual therapy | Not stated | 25 sessions of unstated frequency. Average programme duration was nine months | Monthly reviews of audiotapes from each therapist by a clinical supervisor, who provided feedback on fidelity issues. |

*(Continued)*

**Table 3.** (Continued)

| Study | Name of intervention | Goal of intervention | Materials and procedures | Provider | Mode of delivery | Location delivered | Frequency and duration | Fidelity |
|---|---|---|---|---|---|---|---|---|
| Grabbe et al., 2013 | Shelter-Based Gardening | Improve the mental health of homeless women | Group gardening sessions. Informal garden planning, simple food preparation, nutrition and horticulture education were incorporated into the sessions | Nurses and shelter staff | Group gardening sessions supervised by nurses and shelter staff | Garden in a homeless day shelter | Twice-weekly two-hour drop-in sessions, over a one-year period | Not monitored |
| Graziano et al., 2023 | Parent–child interaction therapy (PCIT) and Child–parent psychotherapy (CPP) | To improve child and parental outcomes | PCIT is a manualized evidence based BPT program that integrates social learning and attachment theories. It is divided into two phases: child-directed interaction (CDI), which resembles traditional play therapy and parent directed interaction (PDI), which resembles clinical behaviour therapy. CPP is a relationship-based treatment that was originally developed to improve the psychological and relational functioning of young children exposed to trauma. | Shelter clinicians | CPP is conducted with the parent–child dyad in unstructured weekly hour-long sessions. | Homeless shelter | 6 weekly sessions | For PCIT, counsellors received weekly supervision by a licensed clinical psychologist, who was a certified trainer by PCIT International. For CPP, a licensed mental health counsellor who had completed CPP training provided biweekly supervision and consultations. |
| Guo et al., 2012 Slesnick & Erdem 2013 | Ecologically-Based treatment (EBT) | Improve health and housing outcomes, and reduce substance misuse and parenting stress among homeless mothers with substance misuse problems, improve child behaviour | Housing in an apartment and three months of rental and utility assistance. Six months of case management and Community Reinforcement Approach (CRA) therapy sessions to address substance misuse | Therapists who had undergone two days of initial training in EBT | Rental and utility support plus face-to-face individual case management and therapy sessions | Not stated | Rental and utility assistance for three months. Up to 20 therapy sessions and 26 case management sessions over a six-month period | 35 therapy sessions were recorded, transcribed, coded and rated for adherence to CRA techniques. Fidelity to CRA techniques and competence rated 'good' |
| Harpaz-Rotem et al., 2011 | Community Residential Treatment | Improve mental and physical health outcomes among homeless female veterans | Housing in a residential treatment unit with provision of social and clinical services | Unit staff and peer counsellors | Temporary housing in the unit plus social and clinical services (no further detail) | Residential treatment units | Participants stayed in residential treatment for at least 30 days | Not monitored |

(*Continued*)

**Table 3.** (Continued)

| Study | Name of intervention | Goal of intervention | Materials and procedures | Provider | Mode of delivery | Location delivered | Frequency and duration | Fidelity |
|---|---|---|---|---|---|---|---|---|
| **Hernandez-Ruiz et al., 2005** 130 | Music Therapy | Reduce anxiety and improve sleep among homeless women in domestic violence shelters | Individual sessions of listening to relaxing music and performing guided progressive muscle relaxation techniques | Researcher | Individual relaxation sessions | Domestic violence shelters | Five 20-minute sessions on consecutive days | Not monitored |
| **Herschell et al., 2017** | Parent-Child Interaction Therapy (PCIT) | Improve parenting practices and mental health among homeless mothers living in domestic violence shelters, improve child behaviour | Individual sessions with therapist, mother and child, aimed at enhancement of the parent-child relationship, effective discipline and boundary-setting. | Therapists, and managers who had completed a year-long training programme in PCIT | Individual sessions | Domestic violence shelters. If women left shelter before programme completion, intervention could be continued in their new home | 12–20 once-weekly one-hour sessions | 42 sessions were videotaped and assessed for fidelity by researchers. 90% of sessions had excellent fidelity to essential PCIT components |
| **Johnson & Zlotnick, 2006** | Helping to Overcome PTSD with Empowerment (HOPE) | Improve symptoms of PTSD and depression and reduce resource loss among homeless women with PTSD or subthreshold PTSD living in domestic violence shelters | Individual CBT sessions covering safety-planning and skills for managing PTSD, substance misuse and depression | Researchers | Individual sessions | Domestic violence shelters | Up to 12 twice-weekly sessions | Not monitored |
| **Johnson et al., 2011** | HOPE | Improve symptoms of PTSD and depression, reduce resource loss and re-abuse and increase social support and empowerment among homeless women with PTSD or subthreshold PTSD living in domestic violence shelters | Individual CBT sessions covering safety-planning and skills for managing PTSD, substance misuse and depression | The first author plus five therapists who had undergone a 12-hour workshop on HOPE | Individual sessions | Domestic violence shelters | Up to 12 1–1.5-hour sessions up to twice-weekly over a maximum of eight weeks | Fidelity was rated by independent therapists for 27 sessions. Providers on average obtained 'excellent' ratings for adherence and competence. |

*(Continued)*

**Table 3.** (Continued)

| Study | Name of intervention | Goal of intervention | Materials and procedures | Provider | Mode of delivery | Location delivered | Frequency and duration | Fidelity |
|---|---|---|---|---|---|---|---|---|
| Johnson et al., 2016 | HOPE | Improve symptoms of PTSD and depression, reduce resource loss and re-abuse and increase social support and empowerment among homeless women with PTSD or subthreshold PTSD living in domestic violence shelters | Individual CBT sessions while resident in the shelter, covering safety-planning and skills for managing PTSD, substance misuse and depression. After shelter exit, therapy sessions in the community and re-evaluation of goals and safety | Four therapists | Individual sessions | Domestic violence shelters, then after shelter exit, in women's homes or community spaces | 16 once-weekly one-hour sessions | Fidelity was rated by independent therapists for 30 sessions. Providers on average obtained 'good' to 'excellent' ratings for adherence and competence. |
| Johnson et al., 2020 | HOPE | Improve symptoms of PTSD and depression, reduce resource loss and re-abuse and increase social support and empowerment among homeless women with PTSD or subthreshold PTSD living in domestic violence shelters | Individual CBT sessions while resident in the shelter, covering safety-planning and skills for managing PTSD, substance misuse and depression. After shelter exit, therapy sessions in the community and re-evaluation of goals and safety | Four therapists who had undergone a two day training workshop | Individual sessions | Domestic violence shelters, then after shelter exit, in women's homes or community spaces | 16 once-weekly one-hour sessions | Fidelity was rated by independent therapists for 34 sessions. Providers on average obtained 'very good' ratings for adherence and competence. |
| Jouriles et al., 2009 | Project Support | Reduce conduct problems in children exposed to domestic violence, and improve mothers' parenting skills and mental health | Individual sessions with mothers and children to a) teach child management skills through didactic instruction, role play, written materials and feedback and b) provide advocacy and support for mothers through needs assessments, safety assessments, emotional support and referrals to community services | Eight therapists who had undergone extensive training in the techniques of the intervention and had passed a mastery test. | Individual sessions with mother and child | The new homes of mothers exiting domestic violence shelters | Weekly sessions for up to eight months | Therapists were closely supervised, sessions were recorded and feedback on fidelity was given in weekly feedback sessions |
| Kahan et al., 2020 | Women aged 16–24 years experiencing homelessness and gender-based violence | 23 | Canada, community resource centre for homeless youth in Toronto | Qualitative study | Multifactorial intervention | - | Acceptability determined through semi-structured interviews | |

*(Continued)*

**Table 3.** (*Continued*)

| Study | Name of intervention | Goal of intervention | Materials and procedures | Provider | Mode of delivery | Location delivered | Frequency and duration | Fidelity |
|---|---|---|---|---|---|---|---|---|
| **Kim & Kim, 2001** | Group Intervention for Battered Women | Decrease depression and anxiety and improve self-esteem of women living in domestic violence shelters | Group therapy which uses teaching and counselling based on a crisis intervention model. Themes covered included assessing trauma, understanding self, managing emotions, identifying batterer characteristics and learning stress management strategies | One of the researchers | Group sessions | Domestic violence shelters | Eight once-weekly 1.5-hour sessions | Not monitored |
| **Lako et al., 2018** | Critical Time Intervention (CTI) | Improve quality of life, self-esteem and social support, prevent re-abuse, and reduce symptoms of depression, PTSD and psychological distress amongst women transitioning from domestic violence shelters into the community | Individual case management which involves needs and strengths assessments, assistance in connecting with community support services and accompaniment to appointments. Eventual transition of responsibility for care needs to community services. | Case managers who have undergone three-day CTI training | Individual case management: face-to-face meetings | Domestic violence shelters initially, then in the community once women have exited shelter | Women are assigned a case manager who is available to them for nine months | Fortnightly meetings with supervisors who review case notes and discuss fidelity. Fidelity reported as 'fair'. |
| **Mallory et al., 2022** | Housing and supportive services | Improving depression symptoms and substance use in homeless mothers. | Provision of independent housing, (Strengths-Based Outreach and Advocacy (SBOA), HIV prevention and substance use/ mental health counselling. | Up to 33 sessions of SBOA addressing basic needs, obtaining government entitlements and connecting mothers to other needed support; 2 sessions of HIV prevention intervention; up to 18 sessions of counselling | Not stated | Not stated | Sessions are to be attended within 6 months from randomisation | Not stated |
| **Marin et al., 2021 Rodriguez-Moreno, 2020** | Unified Protocol for Transdiagnostic Treatment of Emotional Disorders | Improve symptoms of depression and anxiety and increase psychological wellbeing, perceived health and social support among women living in homeless shelters | Group CBT sessions covering motivation, mindfulness, cognitive flexibility, preventing emotional avoidance and maladaptive behaviours, and situation-based emotion-focused exposure. | Two therapists who had worked on adapting the Unified Protocol to a population of homeless women | Group sessions with maximum ten participants | Homeless shelters | Twelve once-weekly 1.5-hour sessions | Periodic meetings with study authors to discuss fidelity, self-assessment of adherence |

(*Continued*)

**Table 3.** (Continued)

| Study | Name of intervention | Goal of intervention | Materials and procedures | Provider | Mode of delivery | Location delivered | Frequency and duration | Fidelity |
|---|---|---|---|---|---|---|---|---|
| **Noh et al., 2018** | Resilience Enhancement Programme | Improve protective factors associated with resilience among runaway female youth living in homeless shelters | Group sessions covering themes of self-esteem, self-regulation, relational skills, problem solving and goal setting. | The primary investigator and a mental health nurse | Group sessions with 4–6 participants | Homeless shelters | Eight twice-weekly 1.5-hour sessions | Not monitored |
| **Nyamathi et al., 1998** | Specialised AIDS Education Programme | Improve AIDS knowledge, emotional wellbeing and coping skills, and reduce risk-taking behaviour and depression among homeless women | Group sessions covering the topics of AIDS aetiology, prevention and testing, as well as decision-making skills, stress reduction, information seeking, and self-esteem enhancement | Nurse and outreach worker | Group sessions | Not stated | Eight once-weekly two-hour sessions, plus two-hour top-up sessions at six- and 12-month follow-up | Not monitored |
| **Nyamathi et al., 2017** | Dialectical Behavioural Therapy–Corrections Modified (DBT-CM) | Improve abstinence from substances in homeless recently incarcerated female parolees and probationers | Group DBT sessions covering topics of eliminating cues and burning bridges to substance use, building a life worth living, observing urges, adaptive denial, alternative rebellion. In addition, individual sessions covering goal setting, chain analysis and solution analysis | Four community health workers and two nurses who received ten days of DBT-CM training | Group sessions with 5–7 participants and individual sessions | Not stated | Six group and six individual sessions of one-hour duration over a 12-week period | Frequent monitoring and rating of fidelity for group and individual sessions |

**Table 3.** (Continued)

| Study | Name of intervention | Goal of intervention | Materials and procedures | Provider | Mode of delivery | Location delivered | Frequency and duration | Fidelity |
|---|---|---|---|---|---|---|---|---|
| **O'Campo et al., 2023** | Housing First | Meet the housing and treatment needs of the chronically homeless population. | Programme developers encourage homeless persons to define their own needs and goals and, if they wish, they can be immediately provided an apartment without any prerequisites for psychiatric treatment or sobriety. People who take part in the intervention are flexible recovery-orientated support by the program's Assertive Community Treatment (ACT) team, but they are not obligated to participate. | Pathways to Housing. ACT is a community-based interdisciplinary team that includes social workers, nurses, psychiatrists and vocational and substance misuse counsellors available to assist 24/7. | Face-to-face | Not stated | Not stated | Not monitored |
| **Rodriguez-Moreno et al., 2022 and 2023** | Unified Protocol for Transdiagnostic Treatment of Emotional Disorders for Homeless Women (UPHW) | Improve symptoms of depression and anxiety and increase psychological wellbeing, perceived health and social support among women living in homeless shelters. | UPHW consists of five core modules: mindfulness emotion awareness, cognitive flexibility, identifying and preventing patterns of emotional avoidance and maladaptive emotion-driven behaviours, increasing awareness and tolerance of emotion-related physical sensations and interceptive and situation-based emotion focused exposure. | Psychology PhD candidates trained in the UPHW protocol, with extensive experience of working with homeless individuals in clinical practice | Group sessions, up to 10 participants | Homeless shelters | 12 weekly face-to-face sessions lasting 1.5h | Periodic meetings to assess adherence. After each sessions therapists completed a questionnaire assessing the degree of adherence to goals, contents and activities. |

(*Continued*)

**Table 3.** (Continued)

| Study | Name of intervention | Goal of intervention | Materials and procedures | Provider | Mode of delivery | Location delivered | Frequency and duration | Fidelity |
|---|---|---|---|---|---|---|---|---|
| Sacks et al., 2004 | Homelessness Prevention Therapeutic Community | Reduce substance misuse, improve parenting skills and mother-child relationships, achieve employment, stabilise housing, develop social supports and achieve reintegration with society among homeless, substance-abusing mothers | Temporary shared housing for homeless mothers and children. Mothers attend parenting skills groups, substance misuse prevention groups, individual and family counselling sessions, work-readiness programmes, re-entry to community groups and housing education groups. Women also receive individual case-management and housing assistance. | Supervisors, counsellors, house managers and peer mentors | A range of programmes for individuals and groups | Therapeutic community housing | Average duration of the programme is not specified | Not monitored |
| Salem et al., 2017 | Frailty Intervention | Improve physical, psychological and social elements of frailty among frail and pre-frail homeless women | Group sessions educating on nutrition, physical activity, social support, stress management, hygiene and bloodborne virus transmission. Individual case management sessions offering assessment and referrals to community services | Two community health workers who underwent six days of training | Group and individual sessions | Homeless day centre | Six group sessions of one-hour duration and six individual sessions of 20-minutes duration over six weeks | Not monitored |
| Samuels et al., 2015 | Family Critical Time Intervention (FCTI) | Support homeless mothers with mental health and substance misuse problems transitioning from shelters to community housing to create the necessary connections with community agencies to support their families, and thus improve their mental health outcomes | Case manager conducts needs assessments and provides links to community resources such as mental health and substance misuse services, employment, child-care and benefits assistance. Case managers work with mothers to secure housing and support mothers with the transition to community living | Case managers who have been trained in FCTI approaches | Individual case management | Homeless shelters initially, then in the community once women have exited shelters | Women are assigned a case manager who is available to them for nine months | Weekly supervision meetings between case managers and supervisors. Fidelity reported as 'good'. |

(Continued)

**Table 3.** (Continued)

| Study | Name of intervention | Goal of intervention | Materials and procedures | Provider | Mode of delivery | Location delivered | Frequency and duration | Fidelity |
|---|---|---|---|---|---|---|---|---|
| Shors et al., 2014 | Mental and Physical Training (MAP) | Improve mental and physical health outcomes in young homeless women | Group focused-attention meditation sessions and group directed exercise sessions | Not stated | Group sessions | Homeless shelter | 16 twice-weekly one-hour meditation and exercise sessions | Not mentioned |
| Slesnick & Erdem, 2012 | EBT | Improve mental health, housing status and employment and reduce substance misuse and stress among homeless mothers with alcohol or substance misuse problems, improve child behaviour | Housing in an apartment and three months of rental and utility assistance. Six months of case management and Community Reinforcement Approach (CRA) therapy sessions to address substance misuse | Three therapists who had completed two days of training in CRA | Individual housing assistance, case management and therapy | Homeless shelters initially, then in the community once women have exited shelters | Three months of rental and utility assistance, six months of case management, six months of therapy | Audio recordings of therapy and case management sessions were reviewed by supervisors |
| Slesnick et al., 2023 | Housing and support services including Community Reinforcement Approach (CRA) | Improve self-efficacy and reduce substance misuse in young homeless mothers. | Independent housing and supportive services including HIV prevention, case management and CRA. | Graduate student therapist | Individual therapy sessions | Not stated | Three months of rental and utility assistance, six months of case management, six months of therapy and two sessions of HIV prevention | CRA sessions were coded by two trained graduate students for adherence and competence. |
| Stahler et al., 2005 Stahler et al., 2007 | Bridges to the Community | Reduce substance misuse, risk-taking behaviours and depression and increase self-esteem in homeless mothers with substance misuse problems residing in a substance misuse rehabilitation programme | Social support through a Community Anchor Person (CAP) who provides mentorship, practical assistance and introductions to community support networks. Group workshops teaching African American culture and dance, Bible studies, and life skills (including parenting skills, healthy relationships and decision-making) | CAP attended three-hour monthly training sessions, not stated who administered group activity | Individual social support and group sessions | Residential substance misuse programme initially, contact with CAP continued in community once housing found | CAPs were required have 20 hours contact per month with women over a one-year period. Workshops were held weekly during residential treatment | Not monitored |

(*Continued*)

**Table 3.** (Continued)

| Study | Name of intervention | Goal of intervention | Materials and procedures | Provider | Mode of delivery | Location delivered | Frequency and duration | Fidelity |
|---|---|---|---|---|---|---|---|---|
| Upshur et al., 2015 | Collaborative Care Model | Reduce alcohol consumption, improve housing stability and health among homeless women who screen positive for alcohol misuse at a primary care provider (PCP) practice. | The electronic medical record (EMR) at participating PCP practices was modified to contain information about alcohol screening results. PCPs were educated on how to intervene with brief motivational interventions, provision of educational materials, alcohol reduction goal-setting and referrals. Care managers provided individual sessions on self-management, goal-setting and linkage to appropriate services | PCPs (doctors, nurse practitioners and physician assistants) attended five days of training. Care managers received 20 hours of training. | Individual sessions with PCPs and care managers | Primary care provider practice | 5–7 appointments with PCPs over six months, 15 appointments with care managers over six months | Weekly supervision meetings between care managers or PCPs and supervisors at which time fidelity was reviewed. Fidelity reported as 'fair'. |
| Weinreb et al., 2016 | Integrated Care Model for Homeless Women | To reduce symptoms of depression in homeless mothers | Depression screening was introduced at the PCP practice and the EMR was modified to contain information about screening results. A care manager was trained to conduct mental health and social assessments, provide psychoeducation, facilitate goal-setting and self-management, and assist with practical needs. The PCP was trained to collaborate with the care manager on outreach, symptom monitoring and treatment modification. | PCPs (doctors) received four hours of initial training, care managers received 20 hours of initial training | Individual sessions with PCPs and care managers | Primary care provider practice | 1–2 appointments with PCP or care manager per week for the first 6–8 weeks, then appointment frequency at discretion of patient and PCP for remainder of the six months | Fidelity was assessed in weekly supervision calls between supervisor and PCPs and care managers. Fidelity reported as 'excellent'. |

improving depressive symptoms in this population, and in most cases, improvements relative to baseline are long-lasting [43–46,51,52,54,55,73,74].

At post-intervention testing, in all studies there was a statistically significant improvement in depression severity in the intervention group compared to baseline. In six of the seven controlled studies, there was a significantly greater improvement in depression severity in the

**Table 4. EPHPP quality ratings for included quantitative studies.**

| Study | Component ratings | | | | | | Overall rating |
|---|---|---|---|---|---|---|---|
| | Selection bias | Study design | Confounders | Blinding | Data collection methods | Withdrawals and drop-outs | |
| Strong | *Representative of target population and 80% participation* | *RCT or CCT* | *Controlled for at least 80% of confounders* | Assessor and participants blinded | Valid and reliable tools | 80% or more at follow-up | No WEAK ratings |
| Moderate | *Likely to be representative and 60–79% participation* | *Cohort analytic, case control or interrupted time series* | *Controlled for 60–70% of confounders* | Assessor or participants blinded | Valid tools but reliability not measured or described | 60–79% at follow up | One WEAK rating |
| Weak | *Not likely to be representative and <60% participation* | *Other designs* | *Controlled for <60% of confounders* | Assessor and participants aware | Validity and reliability not measured or described | <60% at follow up | Two or more WEAR ratings |
| Bain et al., 2014 | Moderate | Strong | Weak | Moderate | Weak | Weak | Weak |
| Bani-Fatemi et al., 2020 | Moderate | Moderate | NA | Moderate | Strong | Strong | Strong |
| Castaños-Cervantes, 2019 | Weak | Strong | Strong | Moderate | Moderate | Strong | Moderate |
| Constantino et al., 2005 | Moderate | Strong | Strong | Moderate | Strong | Strong | Strong |
| Desai et al., 2008 | Moderate | Moderate | Strong | Moderate | Strong | Weak | Moderate |
| Graziano et al., 2023 | Moderate | Strong | Moderate | Moderate | Strong | Weak | Moderate |
| Guo et al., 2016 | Moderate | Strong | Strong | Moderate | Strong | Strong | Strong |
| Harpaz-Rotem et al., 2011 | Moderate | Moderate | Strong | Moderate | Strong | Weak | Moderate |
| Hernandez-Ruiz et al., 2005 | Moderate | Moderate | Moderate | Moderate | Strong | Weak | Moderate |
| Herschell et al., 2017 | Weak | Moderate | NA | Strong | Strong | Weak | Weak |
| Johnson & Zlotnick, 2006 | Moderate | Moderate | NA | Moderate | Strong | Moderate | Strong |
| Johnson et al., 2011 | Moderate | Strong | Strong | Moderate | Strong | Strong | Strong |
| Johnson et al., 2016 | Moderate | Strong | Strong | Moderate | Strong | Moderate | Strong |
| Johnson et al., 2020 | Moderate | Strong | Strong | Strong | Strong | Moderate | Strong |
| Jourilees at al., 2009 | Moderate | Strong | Weak | Strong | Strong | Strong | Moderate |
| Kim & Kim, 2001 | Moderate | Moderate | Weak | Moderate | Strong | Weak | Weak |
| Lako et al., 2018 | Moderate | Strong | Weak | Strong | Strong | Strong | Moderate |
| Mallory et al., 2022 | Moderate | Strong | Strong | Moderate | Strong | Strong | Strong |
| Marin et al., 2021 | Weak | Moderate | NA | Moderate | Weak | Weak | Weak |
| Noh et al., 2018 | Moderate | Moderate | Strong | Moderate | Strong | Strong | Strong |
| Nyamathi et al., 1998 | Moderate | Strong | Strong | Moderate | Strong | Moderate | Strong |
| Nyamathi et al., 2017 | Moderate | Strong | Strong | Moderate | Strong | Strong | Strong |
| O'Campo et al., 2023 | Moderate | Strong | Strong | Moderate | Strong | Strong | Strong |

*(Continued)*

**Table 4.** (Continued)

| | Component ratings | | | | | | |
|---|---|---|---|---|---|---|---|
| Rodriguez-Moreno et al., 2020 | Moderate | Moderate | Strong | Strong | Strong | Weak | Moderate |
| Rodriguez-Moreno et al., 2022 | Moderate | Strong | Moderate | Strong | Strong | Moderate | Moderate |
| Rodriguez-Moreno et al., 2023 | Moderate | Strong | Moderate | Strong | Strong | Moderate | Moderate |
| Sacks et al., 2004 | Moderate | Moderate | Strong | Moderate | Strong | Weak | Moderate |
| Salem et al., 2017 | Moderate | Strong | Weak | Moderate | Strong | Moderate | Moderate |
| Samuels et al., 2015 | Moderate | Strong | Strong | Moderate | Strong | Moderate | Strong |
| Shors et al., 2014 | Moderate | Moderate | Weak | Moderate | Strong | Weak | Weak |
| Slesnick & Erdem, 2012 | Moderate | Moderate | NA | Moderate | Strong | Strong | Strong |
| Slesnick & Erdem, 2013 | Moderate | Strong | Strong | Moderate | Strong | Strong | Strong |
| Slesnick et al., 2023 | Moderate | Strong | Strong | Moderate | Strong | Strong | Strong |
| Stahler et al., 2005 | Moderate | Moderate | Strong | Moderate | Strong | Moderate | Strong |
| Stahler et al., 2007 | Moderate | Strong | Strong | Moderate | Strong | Weak | Moderate |
| Upshur et al., 2015 | Moderate | Strong | Weak | Moderate | Strong | Strong | Moderate |
| Weinreb et al., 2016 | Moderate | Moderate | Strong | Moderate | Strong | Moderate | Strong |

intervention group relative to control between pre- and post-testing [44,45,51,52,54,73,74]. Four of these studies used CBT-based interventions: specifically, the HOPE protocol [44,45], an age-appropriate CBT intervention for young homeless women [51], and a unified CBT protocol for transdiagnostic treatment of multiple mental disorders [52]. The fifth study used a resilience enhancement therapy protocol for young homeless women [54] and the sixth one a Unified Protocol for Transdiagnostic Treatment of Emotional Disorders [73,74].

In seven studies there were additional outcome measurements after the first post-intervention measurement [43–46,51,52,54]. Improvements in depression severity in the psychotherapy group compared to baseline remained significant at all follow-up time points in six of these studies, suggesting that the effects of psychotherapy had good durability [43–46,51,54]. In a single study of the unified CBT protocol, decrease in depression severity from baseline remained significant at three-month follow-up, but was no longer significant at six-month follow-up in the intervention group [52].

**Multifactorial interventions.** Six studies examined the effect of multifactorial interventions on depression severity [41,47,57,61,75,76]. While depression severity in the intervention group improved significantly in over a half of these studies, there was limited evidence that multifactorial interventions were superior to control conditions in improving depression outcomes. In the first post-intervention outcome measurement, depression severity improved relative to baseline in five studies [41,47,57,75,76]. The decrease in depression severity in the intervention group relative to baseline was significant in two studies: one of the EBT studies

and the trial of the AIDS education programme [41,76]. In both of these studies, improvements in depression severity remained significant relative to baseline at 12-month follow-up. Both studies utilised a control group; there was no significant difference between intervention and control groups in change in depression severity over time [41,76].

**Social support interventions.** One controlled study measured the effect of a social support interventions on depression severity, but poor outcome reporting prevents conclusions from being drawn about the intervention's effectiveness [48]. The intervention involved a combination of a support person assigned for one year and culturally sensitive group social activities, with a focus on religious faith. In both control and intervention groups, there was a reduction in depression severity between baseline and 18-month follow-up, but statistical significance was not reported. There was also no measurement of the relative change in depression severity in the intervention group compared to the control group.

**Relaxation and recreation interventions.** One study of a relaxation and recreation intervention demonstrated superiority to control in improving depression severity at immediate post-testing [60]. The intervention involved meditation and directed group exercise for young homeless women. Depression severity significantly improved from pre- to post-testing in the intervention, but not the control group, with significant group x time effects. However, there was no further follow-up testing to determine durability of results.

**Case management interventions.** One controlled study of a case management intervention measured effect upon depression severity and failed to demonstrate any benefit [66].

**Collaborative care model interventions.** One controlled study investigated the effect of a collaborative care model intervention on depression, and while the intervention group experienced significant improvement in their symptoms, there was no evidence that the intervention was superior to control conditions [69].

## Effectiveness of interventions for homeless women in reducing anxiety

**Psychotherapy interventions.** Five controlled studies examined the effect of psychotherapy interventions on anxiety. In all studies, there were greater improvements in anxiety in the intervention group compared to control at one or more time points [51,52,54,55,73,74].

At post-intervention, in four studies there was a statistically significant reduction in anxiety in the intervention group compared to baseline [51,52,55,73,74] while in a fifth study, anxiety reduced, but significance was not reported [54]. In two studies of CBT-based therapies and a study of the Unified Protocol for Transdiagnostic Treatment of Emotional Disorders [73,74] there was also a significantly larger reduction in anxiety in the intervention group relative to the control group between pre- and post-testing [51,52].

In three studies there were additional outcome measurements at 1–6 months post-intervention [51,52,54]. In the study of resilience enhancement therapy, a greater reduction in anxiety score over time in the intervention group compared to the control group was sustained at one-month follow-up, however, significance level was not reported [54]. In the remaining two studies, anxiety score was only measured in the intervention group at time points after post-intervention. In the study of CBT in young homeless women, at two months post-intervention, anxiety scores in the intervention group remained significantly lower than at baseline [51]. In the study of the unified CBT protocol, the difference between baseline and follow-up anxiety severity remained significant at three-month follow-up, but anxiety increased at six months and was no longer significantly different from baseline [52].

**Multifactorial interventions.** One uncontrolled before-and-after study of a multifactorial intervention did not demonstrate any effect of the intervention on anxiety levels over time [75].

**Recreation and relaxation interventions.**   Two studies measured the effect of recreation and relaxation interventions on anxiety, with mixed results [60,65]. The first study of meditation and physical activity intervention demonstrated a statistically significant reduction in anxiety severity from baseline to immediate post-intervention testing, however there was no significant difference in change over time between the intervention and control groups [60]. The second study of relaxation to music [65] showed a significantly larger reduction in anxiety score in the intervention compared to the control group from pre- to post-intervention.

## Effectiveness of interventions for homeless women in reducing PTSD

**Psychotherapy interventions.**   Five studies reported the effects of psychotherapy interventions on PTSD severity [43–46,50]. In all studies, PTSD improved over time in all intervention groups and these improvements were maintained over the follow-up period, but evidence regarding superiority of intervention to control condition was mixed. Four of these studies utilised the CBT-based HOPE protocol, which was targeted specifically at reducing PTSD symptoms in domestic violence survivors [43–46]. The fifth study examined the effects of a different CBT-based programme in homeless female veterans [50].

Only two studies, both evaluating the HOPE protocol [45,46], reported the difference between intervention and control groups at post-intervention and each subsequent follow-up point. In one study PTSD severity was significantly lower in the intervention group compared to the control group at post-intervention and three-month and six-month follow-up [45], while in the second one no significant difference in PTSD severity were found between intervention and control groups at post-intervention or three- or six-month follow-up [46].

**Multifactorial interventions.**   Two studies of different multifactorial interventions utilised PTSD severity as an outcome measure, and neither demonstrated clear evidence of effectiveness [56,75]. In an uncontrolled study of group psychotherapy and recreation for young homeless women, PTSD severity was not significantly lower than baseline at two- and eight-months post-intervention [75]. In a study of psychotherapy and residential housing in a drug treatment facility for homeless female veterans, both intervention and control groups experienced significant decreases in PTSD severity over the study period. However, there was no significant difference in the change in PTSD severity over the 12-month follow-up between intervention and control groups [56].

**Case management interventions.**   One study demonstrated that a case management intervention was superior to control in reducing PTSD severity [66]. In this study of CTI, there were no significant between-group differences in PTSD severity at baseline, but immediately after the nine-month intervention, participants in the intervention group had significantly lower PTSD severity than the control group. There was no further follow-up to determine the durability of this change.

## Effectiveness of interventions for homeless women in reducing substance misuse

**Psychotherapy interventions.**   Two controlled studies of different psychotherapy interventions measured substance misuse, with mixed results concerning effectiveness. In both interventions, psychotherapy was specifically targeted at reducing substance misuse. Over the 12-month period of a CBT-based intervention for homeless female veterans with mental health or substance misuse problems, drug misuse scores, alcohol misuse scores and days of alcohol use in the past month all decreased significantly in both intervention and control groups, with no significant difference in the rate of reduction between intervention and control [50]. By contrast, days of drug use in the past month decreased significantly over the study period in

the control group but not in the intervention group. The findings of a DBT-based programme for homeless female parolees and probationers showed reduced drugs and alcohol use in both intervention and control groups at three-months post-intervention, with a significantly larger increase in abstinence in the intervention group [53].

**Multifactorial interventions.** Ten studies reported the effects of multifactorial interventions on substance misuse, with universal improvements from baseline but mixed evidence that interventions were superior to control [42,47,56–59,61,75–77]. Two of these studies examined the effects of the EBT protocol [47,78]. One study of a homelessness-prevention therapeutic community intervention did not report on individual substance misuse outcome measures, but rather compared a substance misuse domain (a composite of multiple outcome measures) between intervention and control groups [59].

In the nine studies which reported on individual outcome measures, all measures of substance misuse improved in both intervention and control groups over the study period as a whole [42,47,56–58,61,75,76,79]. Six studies reported that these changes were statistically significant [42,47,58,61,76,77]. The interventions in these studies were EBT (housing support, case management and psychotherapy) [47,78], an AIDS education and psychotherapy intervention [76], a frailty prevention and case management programme [77], Housing and support services including Community Reinforcement Approach (CRA) [61] and Pathways to Housing [58].

**Social support interventions.** Two controlled studies of a faith-based social support intervention measured substance misuse severity as an outcome, and while there was some evidence of effectiveness, one study had serious methodological limitations [48,49]. In both studies, all measures of substance use decreased in participants in both intervention and control groups over the follow-up period. In the initial pilot study, 30-day use of alcohol and drugs decreased significantly between baseline and six-months post-intervention in both groups but there was no significant between-group difference in change over time [48].

## Collaborative care model interventions

One controlled study investigated the effect of a collaborative care model intervention on substance misuse, and while outcomes improved over the follow-up period, there was no evidence that the intervention was superior to control [60]. Over the six-month follow-up period, there were significant decreases in 30-day alcohol consumption and significant increases in 30-day alcohol abstinence in both groups, but no significant differences in change in alcohol use over time between the intervention and control groups.

## Effectiveness of interventions for homeless women in improving general psychological health

This section covers all measures of general psychological health status and psychological distress.

**Psychotherapy interventions.** One study of CBT-based psychotherapy for homeless female veterans demonstrated effectiveness in reducing overall psychiatric symptom severity [50]. At three-months post-intervention there was a significant intervention effect and a significant group x time interaction indicating greater improvement in the intervention group compared to control. A study of Unified Protocol for Transdiagnostic Treatment of Emotional Disorders for Homeless Women (UPHW) [73,74] showed significant improvement in emotional functioning between intervention and control group, however, this improvement was not sustained beyond 3 months [73,74].

**Multifactorial interventions.** Eight studies measured the impact of multifactorial interventions on general measures of psychological health status, with mixed evidence of effectiveness [41,47,56,58,59,62,75,76]. Most studies reported significant improvements from baseline in outcome measures, which persisted over time, but in most controlled studies, there was no evidence that the intervention was superior to control conditions.

One study reported only an effect size, rather than individual outcome measurements over time [59]. In the remaining seven studies, there was an improvement over time in measures of general psychological health status in both the control and intervention groups from baseline to final follow-up [41,47,56,58,62,75,76]. In six studies, at final follow-up mental health status either continued to improve [41,47,56] or stabilised [62,75].

Of the six studies with a control group, two reported significantly greater improvements over the study period in at least one measure of psychological health status in the intervention compared to the control group [56,59]. The interventions which were superior to control were a homelessness-prevention therapeutic community for homeless, substance-abusing mothers [59] and residential treatment with psychotherapy for veterans [56].

**Social support interventions.** One RCT of a group-based social support intervention designed to enhance the social resources of women in a domestic violence shelter utilised psychological health status as an outcome measure and demonstrated effectiveness [63]. Psychological distress was not significantly different between control and intervention groups at baseline. At post-intervention, there was a significantly larger reduction in psychological distress in the intervention group compared to the control group. There was no follow-up beyond post-intervention to determine durability of effects.

**Case management interventions.** Two studies of variations on the same time-limited case management intervention (CTI) measured the impact of the intervention on general measures of psychological health, with no evidence of superiority to control [66].

**Parenting interventions.** Three studies of different parenting interventions reported on psychological symptom severity and psychological distress, with limited evidence of effectiveness at immediate post-testing [71,72,80]. In one study of a bonding and reflexivity intervention for sheltered homeless mothers with infants, when psychological distress was measured post-intervention, scores worsened slightly in both control and intervention groups, with no significant differences between groups [71]. By contrast, in an uncontrolled study of a bonding and discipline intervention for homeless victims of domestic violence and their slightly older children (mean age = 4.5 years), there was a significant decrease from baseline to immediate post-testing in number of psychological symptoms and psychological distress [131]. However, only five of the 17 participating parents completed treatment and outcome measurement. There were no additional measurement time points. In a study comparing Parent–child interaction therapy (PCIT) and Child–parent psychotherapy (CPP) [80] significant reduction of parenting stress was observed in both groups, however there was no follow-up to confirm the durability of this effect.

## Acceptability of interventions to improve mental health outcomes in homeless women

**Psychotherapy interventions.** Five quantitative studies reported on the acceptability of psychotherapy interventions to participants [43–46,81]. Across all studies, acceptability of psychotherapy was high.

**Multifactorial interventions.** A single study reported on the acceptability of a multifactorial intervention [82]. This was a qualitative study of participant experiences of the group psychotherapy and recreation intervention for which effectiveness was assessed in a separate

study [75]. Generally, feedback on the intervention was positive. Participants reported that alternating psychotherapy with relaxing recreational activities was helpful, because it enabled them to have a break from the challenging material encountered in therapy sessions. They also raised the importance of women-only sessions in providing a safe space for learning and sharing. Some participants expressed concern that disclosures of personal trauma in psychotherapy sessions could be distressing to other group members with similar trauma histories.

**Social support interventions.** One of the studies of the faith-based social support intervention also reported on acceptability [48]. The Client Satisfaction Score showed that generally, participants found the social support intervention more satisfactory than residential treatment as usual. The intervention group had significantly higher scores for amount of help received, access to desired services, quality of services and competence of staff.

**Recreation and relaxation interventions.** One qualitative study of a recreation intervention investigated acceptability [64]. In this study, a homeless women's day shelter established a community garden with the aim of improving women's mental health. Results showed exclusively positive attitudes towards the intervention. Many participants commented that gardening provided distraction from the trauma of homelessness and improved their mental health. Women commented positively on the sense of community and belonging that developed in the group gardening sessions, which mitigated the marginalisation and social rejection experienced while homeless.

**Collaborative care model interventions.** The study of a collaborative care model intervention for improving substance misuse outcomes also reported the results of a researcher-developed scale for participant satisfaction [60]. Participants in the intervention group were mostly positive about the programme. At six-month follow-up, 69% stated that they were very satisfied with the intervention. Additionally, 73% of participants who improved their substance misuse felt that this could be attributed to the programme. The control group was not surveyed regarding their satisfaction with care.

**Parenting interventions.** The study of a bonding and discipline parenting intervention for women at domestic violence shelters also reported on acceptability [72]. Using the Barriers to Treatment Participation Scale at the mid-treatment point, the authors determined that women experienced substantial barriers to participating in the intervention, but specific barriers were not described. Both completers and non-completers reported high levels of satisfaction at mid-treatment, and at post-treatment, completers continued to report high levels of satisfaction. Women who participated in Parent–child interaction therapy (PCIT) and Child–parent psychotherapy (CPP) [80] reported high levels of overall satisfaction across both interventions noting greatest improvements in their parent–child relationship, feeling like their child made progress in terms of their general behaviour, progress related to their trauma symptoms or traumatic/stressful experiences, and generally positive feelings about the parenting programs. Over 90% indicated that they would likely recommend both programmes to others.

Table 5 below and figures 1–5 (see S4 File) show the number of interventions that produced statistically significant improvement in selected mental health outcomes.

## Discussion

### Summary of findings

The aim of this paper was to systematically review and synthesise evidence on the effectiveness and acceptability of interventions aiming to improve mental health outcomes in homeless women. Thirty-nine studies which evaluated 30 unique interventions were included in the

**Table 5. Effectiveness of interventions on selected mental health outcomes.**

| Mental health outcomes | Depression | | Anxiety | | PTSD | | Substance misuse | | General psychological health status | |
|---|---|---|---|---|---|---|---|---|---|---|
| Intervention | *Sig.* | *NS.* | *Sig.* | *NS.* | *Sig.* | *NS.* | *Sig.* | *NS.* | *Sig.* | *NS.* |
| Psychotherapy | 8 | 0 | 4 | 0 | 5 | 0 | 2 | 0 | 1 | 0 |
| Multifactorial | 2 | 2 | 1 | 0 | 1 | 1 | 6 | 1 | 6 | 1 |
| Social support | 0 | 1 | | | | | 2 | 0 | 1 | 0 |
| Recreation | 1 | 0 | 2 | 0 | | | | | | |
| Case management | 0 | 0 | | | 1 | 0 | | | 1 | 1 |
| CCM | 1 | 0 | | | | | | | | |
| Parenting | | | | | | | | | 1 | 1 |

[a]Intervention group shows significant improvement in ≥1 measure of psychological health status from baseline to ≥1 subsequent time points (Sig., Significant; NS., Not significant)

review. Most studies were conducted in the USA from 2011 onwards, and the most common study design was an RCT. Most studies included were of high or moderate quality.

This review provided moderate evidence that interventions are effective at improving mental health outcomes in homeless women. Half of the controlled studies measuring effectiveness demonstrated that interventions were more effective than treatment as usual in improving at least one mental health outcome. In all but one of the remaining controlled studies, while interventions were not proven to be more effective than control conditions, there was a significant improvement in at least one mental health outcome in both groups from baseline to post-intervention. In three of the four uncontrolled studies measuring effectiveness, there was a significant improvement in at least one mental health outcome from baseline to post- intervention. In approximately two-thirds of studies measuring effectiveness, outcomes were measured at additional time points after the immediate post-intervention period. The longest follow-up was 24-months post-baseline [62].

Outcome measures fell into five categories: depression, anxiety, PTSD, substance misuse and general psychological status. Most studies measured the effect of an intervention on multiple outcomes, and interventions were frequently effective for one outcome and not another. Interventions fell into one of seven categories: psychotherapy, multifactorial, social support, recreation and relaxation, case management, collaborative care and parenting. Significantly more interventions fell into the psychotherapy and multifactorial categories than any other category. Because of the substantial heterogeneity in the nature of interventions within each category, it is difficult to draw definitive conclusions about the effectiveness of each intervention category in improving mental health outcomes. Furthermore, there was significant heterogeneity in participant populations which makes generalisation difficult.

## Comparison to existing literature

In this review, psychotherapy interventions were supported by the strongest evidence of effectiveness. In all studies of psychotherapy interventions, there was improvement in the intervention group relative to baseline in one or more outcome measures. Several of the studies with a control group also demonstrated greater improvement in one or more outcomes relative to control [44,45,50–55,73,74]. The psychotherapy interventions which showed superiority to control in this review were based on CBT [44,45,50–52], DBT [53], crisis intervention model [55], and resilience enhancement [54] protocols. These findings are consistent with a large

body of literature demonstrating that psychotherapies are effective at treating these mental disorders in domiciled populations [83–86]. In most studies in general populations, different psychotherapy protocols demonstrate similar efficacy, with no protocol unequivocally superior [87,88]. Improvements in outcome measures remained significant relative to baseline at 1–12 months post-intake in all studies of psychotherapy but one [52]. This is consistent with the literature on the durability of psychotherapy interventions, which have been shown to produce stable improvements in depression, anxiety and PTSD, and in some cases, ongoing improvement, for up to 20 months [87,89–91]. The reason for the durability of psychotherapy effects is not fully understood, but may relate to the fact that therapy provides participants with a cognitive skillset that can be adapted to cope with future challenges and adversity [86]. The evidence in this review is also consistent with literature on the effectiveness of psychotherapy in mixed-gender homeless populations. A range of psychotherapy protocols have been shown to improve depression, anxiety and substance misuse in mixed-gender homeless populations, including CBT, motivational interviewing, strengths-based therapy, interpersonal psychotherapy, and family therapy [92–98].

Multifactorial interventions also featured prominently in this review, however evidence for their effectiveness was mixed. These interventions were highly heterogeneous, and it was difficult to disentangle the effects of different intervention components on mental health outcomes. Most studies used substance misuse or general psychological health status as an outcome measure. While in the majority of studies, outcomes improved from baseline to follow-up, in most cases where a control group was used, intervention and control groups improved at a similar rate. Studies of six interventions demonstrated superiority to both control and baseline at post-intervention [56–59,61,76].

Housing (either independent, supported or in a therapeutic environment) was a frequent component of multifactorial interventions in this review [41,42,47,56–59,61]. There is a large body of literature on the effectiveness of housing interventions in improving mental health outcomes in general homeless populations. In these interventions, housing is usually supplemented with additional services such as case management and psychotherapy. A distinction is often drawn between Treatment First (TF) and Housing First (HF) approaches. In a TF approach, homeless people must meet criteria to demonstrate that they are ready to maintain stable housing [99,100]. This may involve a period of psychiatric or substance misuse treatment prior to being housed, and ongoing entitlement to housing may be contingent on abstinence from substances or attending mental health treatment. In an HF approach, housing is provided alongside case management or other supports without extensive prior assessment of readiness, or contingency on desired behaviours such as substance abstinence [57,61,99,100]. HF is based on principles of harm-reduction, recognising that for individuals who continue to misuse substances, the harms of homelessness can still be mitigated by provision of housing [101]. Both in this review and in the general literature on homelessness, there is mixed evidence for the effectiveness of multifactorial interventions with a housing component in improving mental health.

Studies of TF interventions in general homeless populations have frequently failed to demonstrate superiority to usual community services in improving mental health outcomes. Two studies of case-management plus abstinence-contingent housing in mixed-gender homeless populations found that while substance misuse improved from baseline to follow- up at 10–12 months, the intervention group did not improve significantly more than those who received case management alone or usual community services [102,103]. In this review, the two studies of interventions with an abstinence-contingent housing component showed mixed results for substance misuse and positive results for psychological wellbeing [56,59]. However, in these studies, the housing was shared accommodation in a therapeutic community [59], and

accommodation in a residential treatment unit [56], which may produce different outcomes compared to independent abstinence-contingent housing.

HF programmes are effective in improving time spent housed, quality of life, hospitalisations and emergency department visits in general homeless populations [57,61,99,101,104–107]. However, there is conflicting evidence as to whether HF programmes improve mental health outcomes. Some studies comparing HF interventions with TF interventions and usual community services have generated positive results, with the HF group demonstrating significantly greater reductions in substance misuse [57,61,100,108,109] and time spent in psychiatric institutions [110]. However, in other studies HF interventions were not shown to be superior to TF or usual community services in terms of improving psychiatric symptom severity and substance misuse [101,106,107,111]. Consistent with the general literature, studies in this review of an HF protocol called EBT failed to demonstrate greater improvements in depression, substance misuse and psychological health status than usual shelter services [41,78]. It is surprising that HF is not more effective at improving mental health outcomes in homeless women, as it addresses the fundamental problem of homelessness: a lack of stable housing. One possible explanation is that because homeless women have usually experienced significant trauma and exclusion, provision of housing will be inadequate to significantly improve their mental status. Additional intervention components such as intensive psychotherapy, parenting skills training and organised social support may be required [104]. Furthermore, newly housed homeless people often experience increased loneliness and isolation when removed from shelter or street social networks, which may limit improvements in mental health [112].

This review generated less evidence for the effectiveness of other intervention categories. Social support interventions produced improvements in substance misuse and general psychological status, but not depression severity [48,49,63]. The general literature on homelessness contained very few additional studies of social support interventions. However, there is extensive evidence that stronger social support networks in homeless people improve mental, physical and social outcomes [34,113,114].

Recreation and relaxation interventions in this review were effective at reducing symptoms of depression and anxiety [60,65]. Several studies in the broader literature on homelessness assessed a diverse range of recreation and relaxation interventions [115–119], generally involving some form of physical activity or creative art. In several studies, participants reported reductions in negative emotion and stress post-intervention [116,118,119]. In some qualitative studies, participants also noted that engaging in new recreational activities fostered a sense of social connectedness [115,117], introduced ways of having fun without substance misuse [115,117] and increased self-esteem [115–117].

In the present review, case management was effective at reducing PTSD symptoms, but not depressive symptoms, and produced mixed effects on general psychological status [66,67]. A recent systematic review of case management in general homeless populations also yielded mixed results [120]. The authors found that case management was more effective at reducing substance misuse than improving other mental health outcomes such as depression. In another review, CTI in mixed-gender homeless populations demonstrated greater efficacy in improving mental health outcomes than other case management protocols such as Intensive Case Management and Assertive Community Treatment [121].

When a collaborative care model was adapted to target depression and substance misuse, there was improvement relative to baseline, but no superiority to usual care in either study [68,69]. There is limited literature on the effect of collaborative care on mental health outcomes in general homeless populations, but a recent systematic review showed that integration of primary care has positive impacts on housing, healthcare utilisation and patient satisfaction for homeless people [122].

Parenting interventions resulted in an improvement in psychological symptoms relative to baseline in two studies, and deterioration of symptoms in another study [71,72,80]. There is a small body of literature evaluating parenting interventions in homeless populations, most studies are methodologically weak, and outcomes generally relate to child behaviour and parenting skills, as opposed to parental mental health. In a review of parenting interventions in homeless shelters, the authors concluded that interventions are effective in helping parents gain knowledge of parenting skills, but may not improve parental or child behaviours [123]. One study of a shelter-based intervention found that while parenting practices and child behaviour improved post-intervention, parental wellbeing did not [124].

Interestingly, in most controlled effectiveness studies, mental health outcomes improved significantly in the control group, and often the rate of improvement was not significantly different between the intervention and control groups. There are various explanations for why this is the case. In many studies, control groups received shelter or treatment facility services as usual, and were often provided with a rich package of services, including referrals, psychotherapy and case management [44–46,48,49,52,59,66,67]. As a result, in some studies, the services received by women in the two groups were not substantially different, a problem which has been observed in studies in general homeless populations [125]. A 'Hawthorne Effect' may also occur, where service providers such as shelters improve the care offered to control participants in response to their awareness of being monitored [125]. In many studies, baseline outcome measurements were taken at shelter entry, often a time of crisis in women's lives, when they were fleeing a violent partner or experiencing rooflessness. There is evidence that women's mental health tends to improve from initial shelter measurements to subsequent time points, regardless of further interventions, and it has been hypothesised that this is due to the removal of an acute stressor such as a violent partner [126–128]. Furthermore, meta-analytic evidence suggests that in general populations, depression and PTSD severity trend downwards in most individuals over time, even in the absence of any treatment [129,130].

Another interesting observation was that the effect of interventions targeting substance misuse diminished over time resulting in an increase in substance use at final follow-up [42,47,75,76], suggesting that durability was a problem for these type of interventions. This phenomenon has also been observed in studies conducted in mixed-gender homeless populations. In these studies, which cover a range of interventions, substance misuse decreased immediately post- intervention but then began to increase again at 3–24 months post-intervention [78,131–134]. In one study, cocaine use returned to baseline at six-month follow-up [133]. This can be explained by the fact that substance misuse is a complex problem where relapse post- intervention is common even in domiciled populations, and homeless individuals have fewer social and economic resources to assist them in maintaining abstinence [132]. Consistent with the findings of this review, in the broader literature on homelessness, post-intervention improvement in other mental health outcomes such as depression and psychological distress generally remained stable or continued to improve at final follow-up [94,131,135].

Acceptability of interventions in all categories was high. Most studies which reported acceptability utilised quantitative surveys which provided limited information. However, the two qualitative studies shed light on some factors which increase acceptability. It was important to women that barriers to entry were addressed, for example through provision of transport. Women preferred interventions delivered in female-only environments, and valued the incorporation of relaxation elements into interventions [64,82]. In one study, women also highlighted the risk of distress to others in group settings where trauma disclosures occur [82]. This is important, as many of the interventions occurred in group settings and had a focus on addressing trauma. The broader literature on the acceptability of health interventions to homeless women reflects some of these concerns, and also raises other barriers to participation.

Three studies of health-seeking behaviours in homeless women identified lack of transport and perceived lack of respect from providers as barriers to participation in health interventions [136–138]. One of these studies also found that health was not a priority for most homeless women, who were more concerned with immediate survival needs [136]. A systematic review of acceptability factors for interventions in homeless populations found that women valued female-only interventions as they felt unsafe in settings where men were present [139]. The review also found that interventions with a parenting or social support component were highly acceptable to homeless women as they fostered feelings of social connectedness and competency as a mother. Interventions with a housing component were often unacceptable to women because they felt that the neighbourhoods in which they were housed were unsafe for their children. This may contribute to the underwhelming results of housing-based interventions in the present review.

## Quality of included studies

A major strength of the studies in this review was the fact that many included long-term follow-up. Thirty studies had additional time points after the initial post-intervention outcome measurement, and the longest duration of follow-up was 24-months post-baseline. This enables conclusions to be drawn about the durability of intervention effects over time. Study design was also a strength, with a large number of RCTs with high internal validity.

Selection bias in most studies was moderate; participants were approached for recruitment in shelters and community homeless resource centres. This is pragmatic, but may miss some of the most severely marginalised homeless women who do not access these resources. Participation rates were high in studies which reported them, with greater than 80% of women agreeing to take part in most studies. Most studies did not report on whether measures were put in place to blind participants to their intervention or control status, or whether participants were aware of the research questions. However, as many interventions took place in shelters or resource centres, which are social spaces, it would be difficult to prevent participants from discussing the study and determining which group they had been assigned to. Some studies tried to avoid contamination by using cluster randomisation or non-random assignment of clinics or shelters to a treatment condition [59,68,69].

A weakness of many studies was the failure to adequately correct for significant differences between intervention and control groups at baseline, resulting in confounding. Furthermore, loss to follow-up was often considerable, with more than 60% of participants dropping out of eight studies, and loss to follow-up not reported in a further two studies. This is difficult to avoid due to the itinerant nature of homeless populations. Multiple testing was very common across studies, and there was often no attempt to control for the effects of multiple testing. This increases the probability that some of the positive findings in the studies were due to random sampling error rather than a true effect [140]. Several studies had small numbers of participants, which could result in studies being underpowered to detect significant effects, and therefore increase Type II error [141]. A few studies did use large samples across a number of different settings, such as different shelters [50,66]. While this may increase generalisability, it frequently meant that the content of the intervention and control varied substantially between settings, and fidelity to the intended intervention was reduced.

## Strengths and limitations

This is the first known systematic review to synthesise evidence on the effectiveness and acceptability of interventions in improving mental health outcomes in homeless women. A strength is that the broad inclusion criteria for interventions enabled examination of the effects

of varied interventions on mental health outcomes, thus acknowledging the wider determinants of mental health. Inclusion criteria for participants were also broad, capturing not only the most visible street-dwelling homeless women, but also women experiencing 'hidden homelessness'. As a result, the review population reflected the diversity of homeless female populations in high- and middle-income countries. Inclusion of a review question relating to acceptability was also a strength, as we wished to incorporate the voices of homeless women themselves into this review; particularly important as this is a highly marginalised population. Another strength of this review is that the majority of studies were of moderate or high quality according to EPHPP criteria [38], and a substantial proportion of methodologically strong RCTs were included.

However, this review does have several limitations. Grey literature was excluded, so findings may overestimate the effects of interventions on mental health outcomes due to publication bias. Furthermore, studies which were not available in English were excluded. While the broad inclusion criteria for both interventions and participants resulted in a comprehensive evidence synthesis, the resulting heterogeneity made categorisation of interventions and comparison of results challenging. Due to heterogeneity in interventions, it was challenging to fit interventions into meaningful categories to which summaries could be applied. This was particularly apparent in the case of multifactorial interventions, which were very diverse. It was also difficult to disentangle the effects of the various components of multifactorial interventions. As a result of the heterogeneity in study design, interventions, outcome measures and participant characteristics, it was not possible to conduct a meta-analysis. Due to the large number of risk and vulnerability factors that cause and perpetuate homelessness, interventions to improve the mental health of homeless women are often complex. To categorise interventions and develop a narrative summary, it was necessary to simplify such complex interventions, to the extent that detail relevant to practical application may not be apparent in this review. Therefore, policymakers and practitioners interested in applying an intervention based on the findings of this review should consult the original papers to determine whether the intervention is likely to be transferrable to their specific context.

## Conclusions and implications for future research and practice

This review highlighted some important gaps in the literature which should be addressed in future research. Certain subgroups of homeless women were under-represented in this review. While a reasonably large number of studies related to interventions for homeless mothers and homeless women in domestic violence shelters, there was very limited literature on homeless women without children and elderly homeless women. Homeless pregnant women and mothers are particularly vulnerable and prone to exploitation since their primary concern is to ensure that their children are safe and have a roof over their heads [7,12]. Constant pressure and stress is likely to have a negative impact on women's mental health, but also on their children's, thus future research should focus on interventions to support both mothers and children experiencing homelessness. Elderly homeless women are also likely to have unique vulnerabilities and needs, and therefore specific research should be directed at determining which interventions are most effective for improving their mental health. No studies used self-harm or suicide as outcome measures. Because homeless women are at substantially higher risk of dying by suicide than women in the general population [142], there is an urgent need for research to determine which interventions are most effective at reducing suicidality in this group. Certain intervention categories such as social support interventions and case management showed promise in improving mental health outcomes, but due to the limited number of studies, it is difficult to draw definitive

conclusions. More research on interventions of this nature is needed to develop a stronger evidence base for their effectiveness.

While several studies measured acceptability as an outcome, most used a single quantitative measure of satisfaction or usefulness, which does not give detailed information about elements of interventions that women found helpful, elements that women found unsatisfactory, and barriers to access. Future RCTs would be enriched by parallel qualitative studies of interventions, which would provide greater insight into their strengths and weaknesses from the perspective of the service user [143]. In the development of future interventions, there would be benefit in involving homeless women in process evaluations, to maximise acceptability of the final intervention and to provide a platform for the empowerment of participants.

Although more research in this field is required, some recommendations for practice can be made based on this review. Psychotherapy interventions had the greatest evidence of long-lasting effectiveness in improving mental health outcomes in homeless women. Therefore, provision of individual or group therapy in residential and community settings is recommended. Although provision of housing as part of a multifactorial intervention showed limited effectiveness in improving mental health outcomes, other studies have demonstrated improvements in other important outcome measures, such as housing stability. When implementing an intervention that includes housing, practitioners should consider adding additional services, such as psychotherapy, to ensure that women's mental health is adequately supported. While other intervention categories in this study showed promise, there was insufficient evidence of effectiveness to make practice recommendations.

## Supporting information

**S1 Checklist. PRISMA 2020 checklist.**
(DOCX)

**S1 File. Full search strategies.**
(PDF)

**S2 File. Data extraction table.**
(PDF)

**S3 File. Detailed characteristics of included studies.**
(PDF)

**S4 File. Effectiveness of interventions figures.**
(PDF)

## Author Contributions

**Conceptualization:** Joanna Anderson, Charlotte Trevella, Anne-Marie Burn.

**Formal analysis:** Joanna Anderson, Charlotte Trevella, Anne-Marie Burn.

**Methodology:** Joanna Anderson, Anne-Marie Burn.

**Project administration:** Charlotte Trevella.

**Supervision:** Joanna Anderson, Anne-Marie Burn.

**Writing – original draft:** Charlotte Trevella.

**Writing – review & editing:** Joanna Anderson, Anne-Marie Burn.

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
