## [Decision Letter · Decision Letter 0]

19 Dec 2023

PONE-D-23-33329Interventions to improve the mental health of women experiencing homelessness: A systematic review of the literaturePLOS ONE

Dear Dr. Burn,

Thank you for submitting your manuscript to PLOS ONE. After careful consideration, we feel that it has merit but does not fully meet PLOS ONE’s publication criteria as it currently stands. Therefore, we invite you to submit a revised version of the manuscript that addresses the points raised during the review process. Please submit your revised manuscript by Feb 02 2024 11:59PM. If you will need more time than this to complete your revisions, please reply to this message or contact the journal office at plosone@plos.org. Please include the following items when submitting your revised manuscript:A rebuttal letter that responds to each point raised by the academic editor and reviewer(s). You should upload this letter as a separate file labeled 'Response to Reviewers'.A marked-up copy of your manuscript that highlights changes made to the original version. You should upload this as a separate file labeled 'Revised Manuscript with Track Changes'.An unmarked version of your revised paper without tracked changes. You should upload this as a separate file labeled 'Manuscript'.If applicable, we recommend that you deposit your laboratory protocols in protocols.io to enhance the reproducibility of your results. Protocols.io assigns your protocol its own identifier (DOI) so that it can be cited independently in the future. For instructions see: https://journals.plos.org/plosone/s/submission-guidelines#loc-laboratory-protocols. Additionally, PLOS ONE offers an option for publishing peer-reviewed Lab Protocol articles, which describe protocols hosted on protocols.io. Read more information on sharing protocols at https://plos.org/protocols?utm_medium=editorial-email&utm_source=authorletters&utm_campaign=protocols.

We look forward to receiving your revised manuscript.

Kind regards,

Ankur Srivastava, Ph.D.

Academic Editor

PLOS ONE

Journal Requirements:

2. We note that your Data Availability Statement is currently as follows: [ll relevant data are within the manuscript and its Supporting Information files.]

Reviewers' comments:

Reviewer's Responses to Questions

**Comments to the Author**

1. Is the manuscript technically sound, and do the data support the conclusions?

Reviewer #1: Yes

Reviewer #2: Yes

2. Has the statistical analysis been performed appropriately and rigorously? 

Reviewer #1: N/A

Reviewer #2: N/A

3. Have the authors made all data underlying the findings in their manuscript fully available?

Reviewer #1: Yes

Reviewer #2: Yes

4. Is the manuscript presented in an intelligible fashion and written in standard English?

Reviewer #1: Yes

Reviewer #2: Yes

5. Review Comments to the Author

Reviewer #1: Review of PONE-D-23-33329

This systematic review examines the state of the literature focused on reducing mental health disparities among women experiencing homelessness. The review appears to have been conducted with rigor and authors offer a thorough account of their processes. Findings are well organized and effective for readers. Overall, the review is effective and offers a helpful contribution to the literature in its summation. I have a few minor comments listed below.

• Unless I missed it, I saw no date range included in the inclusion criteria. Articles were published between 1998 and 2023, but what range was specified in the searches?

• Inclusion criteria specified studies were to consist of >90% girls/women. How many were focused solely on the population of interest and how many were a mixture of genders, where women were >90% of the sample?

• Table 4 is very effective in communicating the qualities of the studies. However, I am interested in the definitions of the classifications in the EPHPP Quality Assessment Tool. For example, what indicates a study has moderate selection bias? Is there a way to effectively communicate these definitions?

• The structure of the results is quite effective in communicating findings. Most readers will likely be interested in identifying a best practice for a specific condition in this population and authors have effectively done that.

Reviewer #2: Summary: This systematic review aimed to examine the effectiveness and acceptability of interventions aimed to improve the mental health of homeless girls and women.

Abstract: clear and concise.

Introduction: A brief description of ‘rough sleepers’ on page 3 may help as I do not believe this term is used outside the UK. I had to look it up.

Overall, introduction is clear and concise. It makes a strong argument for the systematic review in order to address this important issue.

Aims and research questions: Research question 1: as a review hasn’t been done like this before, simply listing interventions would be interesting. In your results you categorize the interventions – that could be a part of your research question or a separate research question.

Aims and Research Questions: The word ‘question’ is missing the ‘s’

Methods: appear appropriate

Search Strategy and selection criteria: a short sentence on what the PICO framework is and regularly used for would be helpful.

Inclusion Criteria: clarify whether any age was included (it appears so but just be explicit)

Study selection: in those articles that were double reviewed, how much agreement/disagreement was there? I don't believe an exact percent agreement necessary but a broad statement would be helpful.

Pg. 11 Psychotherapy interventions; reword/phrase the last two sentences, they read poorly. Possible alternative: “In all studies, psychotherapy was delivered in person, whether in a group context (n=6), individual (n=4) or a mixture of group and individual sessions (n=1). The most commonly use therapeutic model was cognitive behavioral therapy (CBT). In addition, dialectical behavioral therapy, resilience enhancement therapy, and crisis intervention were used."

Study quality: more detail on the CASP checklist needed; what was the criteria – possibly present in a table

Results - if you are able to come up with a table to represent the effectiveness of the studies, that may be helpful to summarize your findings. If you could develop a table that doesn't take up more than a page I would add it, but if not, it probably wouldn't add anything to your article. The way you present them are clear but hard to get a good overall sense of what you found.

Top of pg. 35 – references in last paragraph need fixing

Top of pg. 37 - 39 – extra line between paragraphs in a number of places

Discussion:

Pg. 44 – first sentence of second paragraph. Saying that substance misuse ‘deteriorated’ is not clear to me, I had to re-read this sentence a few times. Reword to make clearer. I believe you are saying that effectiveness of substance misuse programs waned over time with an increase in substance use in follow-up

Conclusions: You mention that homeless women without children and elderly homeless women have unique vulnerabilities and needs that should be addressed, which is certainly true, but I think a particularly unique issue some women face is those who may be pregnant or mothers – they have the added stress of worrying about the child(ren), may face more difficulties in securing adequate housing, and may be more likely to stay in unsafe environments in order to keep a roof over their child(ren). This is very briefly mentioned in the introduction but a few sentences either added to the introduction or conclusions would strengthen your article.

Why do you think only studies from US and Canada (except one) recorded ethnicity?

6. PLOS authors have the option to publish the peer review history of their article (what does this mean?). If published, this will include your full peer review and any attached files.

Reviewer #1: No

Reviewer #2: No

---

## [Author Response · Author response to Decision Letter 0]

5 Jan 2024

Please see our response to reviewers in the uploaded file (Response to reviewers_PONE-D-23-33329_050124)

---

## [Editor Report · Decision Letter 1]

15 Jan 2024

Interventions to improve the mental health of women experiencing homelessness: A systematic review of the literature

PONE-D-23-33329R1

Dear Dr. Burn,

We’re pleased to inform you that your manuscript has been judged scientifically suitable for publication and will be formally accepted for publication once it meets all outstanding technical requirements.

Kind regards,

Ankur Srivastava, Ph.D.

Academic Editor

PLOS ONE

---

## [Editor Report · Acceptance letter]

29 Jan 2024

PONE-D-23-33329R1 

PLOS ONE

Dear Dr. Burn, 

I'm pleased to inform you that your manuscript has been deemed suitable for publication in PLOS ONE. Congratulations! Your manuscript is now being handed over to our production team.

Kind regards, 

on behalf of

Dr. Ankur Srivastava 

Academic Editor

PLOS ONE